# Can We Safely Obtain Formal Oxidation States from Centroids of Localized Orbitals?

**DOI:** 10.3390/molecules25010234

**Published:** 2020-01-06

**Authors:** Martí Gimferrer, Gerard Comas-Vilà, Pedro Salvador

**Affiliations:** Departament de Química and Institut de Química Computacional i Catàlisi, Universitat de Girona, Maria Aurèlia Capmany 69, 17003 Girona, Spain; mgimferrer18@gmail.com (M.G.); gerard.comas7@gmail.com (G.C.-V.)

**Keywords:** oxidation states, localized orbitals, metal carbenes, wavefunction analysis

## Abstract

The use of centroids of localized orbitals as a method to derive oxidation states (OS) from first-principles is critically analyzed. We explore the performance of the closest-atom distance criterion to assign electrons for a number of challenging systems, including high-valent transition metal compounds, π-adducts, and transition metal (TM) carbenes. Here, we also introduce a mixed approach that combines the position of the centroids with Bader’s atomic basins as an alternative criterion for electron assignment. The closest-atom criterion performs reasonably well for the challenging systems, but wrongly considers O-H and N-H bonds as hydrides. The new criterion fixes this problem, but underperforms in the case of TM carbenes. Moreover, the OS assignment in dubious cases exhibit undesirable dependence on the particular choice for orbital localization.

## 1. Introduction

Heuristic concepts play a key role in chemical knowledge. By the time quantum mechanics was readily applicable to chemical systems, there was already a vast amount of chemical information that had been gathered and systematized over decades according to some constructs. It should not be a surprise that most of these chemical concepts are not observable in a strict quantum mechanical sense (hence, they have no few detractors claiming its inherent arbitrariness). Still, they have undoubtedly proven very useful for shedding light into chemical phenomena and more importantly, for achieving true predictions (i.e., without actually performing an experiment or even a computational exercise). Often, the problem of a chemical concept arises when it comes to its quantification as recently stressed by Grunenberg [1]. For instance, there are a myriad of aromaticity indicators or different realizations to compute bond orders that may sometimes lead to different interpretations emerging from the same input. There is, however, a concept of utmost relevance in chemistry, whose flaw was actually the lack of a well-established definition, namely the oxidation state (OS). 

For years, the assignment of oxidation states was performed according to a set of “agreed upon” rules, but no explicit definition of the concept was given. Recently, the entry of OS on IUPAC’s Gold Book has been modified to include a new generic definition following the recommendations of a task group lead by Prof. Karen [2,3]. The current OS definition reads “the atom’s charge after ionic approximation of its heteronuclear bonds”, while bonds between atoms of the same element must always be divided equally. In addition, back-of-the-envelope algorithms applicable to molecules and solids were provided. For instance, in the case of molecular systems, the so-called algorithm of assigning bonds starts by establishing the appropriate Lewis structure of the molecule. Then, the electron pairs between two bonded atoms are assigned to the more electronegative one according to Allen’s scale [4], which represents the easiest application of the ionic approximation. Such a simple recipe works pretty well in most cases. 

The current definition of OS is much more satisfactory than the previous set of agreed upon rules, and clearly represents an improvement. Moreover, a number of spectroscopic techniques and detailed information about the geometrical arrangement of the atoms are indirect experimental probes of the oxidation states (in the former case, referred as spectroscopic oxidation state). Yet, several recent works have exposed some intrinsic limitations of the ionic approximation [5,6]. Postils et al. [5] concluded that, in π-adducts, the local spin state of the π-system determines its formal charge, and concomitantly that of the TM. The cycloheptatryenil (C_7_H_7_) moiety can be found on his π-system as formally (+1) or (−3), fulfilling Hückel’s rule, but also (−1) if it appears in triplet state, following instead Baird’s rule. The case of transition metal (TM) carbenes is also illustrative [7]. Since carbon is more electronegative than the TM, the carbene moiety should keep all four electrons involved in the double-bond, reaching a formal charge of (−2). Thus, the straight application of IUPAC’s rule naturally accounts for the nucleophilic Schrock-type carbenes. On the contrary, Fischer-type carbenes are typically depicted by a carbene unit keeping the σ electron pair and the TM keeping the π electron pair. Notice that such view cannot be reconciled with IUPAC’s winner-takes-it-all rule, so that discerning Fischer or Schrock character from an OS perspective calls for approaches beyond IUPAC’s ionic approximation. 

Most of the ambiguities and caveats in IUPAC’s scheme originate from the inability of atomic electronegativity scales to account for the different chemical environment of atoms within the molecule. On the other hand, the OS must be connected to the electron distribution around the atoms, which can be pretty well described with contemporary electronic structure methods. In our opinion, computational chemistry can and should have a major role in establishing oxidation states, particularly in those difficult cases where subtle details of the electronic structure are most relevant. While it is well-known that partial atomic charges do not match with OS (particularly for high-valent species), there is a generalized misconception that they still represent a sort of non-integer version of OS. One can still find vivid discussions in the literature on the basis of partial atomic charges or atomic spin populations computed one way or another [8,9,10,11,12]. 

When extracting chemical information from wavefunction analysis, one should clearly distinguish the function that is scrutinized from the method chosen to perform the analysis. The latter, in most analyses, refers to how atoms are identified within the molecule, which is essential in the assignation of the OS. In the so-called Hilbert-space analyses, the linear combination of atomic orbital approach to generate the molecular orbitals (LCAO-MO) framework is exploited to collect atomic contributions. When the density function is the one-electron density ρ(**r**) and the atomic orbitals used are those of the underlying one-electron basis set, the well-known Mulliken [13] population analysis is recovered. Orthonormalized atomic orbitals such as those provided by Löwdin orthogonalization [14], Weinhold’s natural orbitals [15], or Ruedenberg’s quasiatomic orbitals [16] lead to more robust atomic populations (i.e., less dependent on the one-electron basis set choice). On the other hand, in real-space analyses the atoms are identified by a region of the three-dimensional physical space, that may be disjoint like in Bader’s quantum theory of atoms in molecules (QTAIM) [17], or overlapping, like in the different flavors of Hirshfeld-type approaches [18]. Real-space analyses are very robust with respect to basis set but still bear the burden of arbitrariness. We do not consider one approach conceptually better than another. In fact, a link has been found between Hilbert-space and real-space analyses by means of a particular set of atomic orbitals, the so-called Mayer’s effective atomic orbitals (eff-AOs) [19,20]. The recipe is simple: (i) Pick a real-space atomic definition such as Hirshfeld’s approach, and obtain the corresponding numerical eff-AOs in that framework; (ii) Expand the original wave function in terms of these eff-AOs and perform a Mulliken-type analysis of the electron density. The original Hirshfeld’s atomic populations will be exactly recovered [20,21]. Such a numerical exercise evidences that there is nothing fundamentally flawed with Mulliken’s approach, it all depends on the underlying basis set used for the analysis. On the other hand, partial atomic charges account by definition for the average number of electrons associated to each atom, while by its actual definition the OS is not any average quantity. In our opinion, one does not need a more suitable or tailored atom-in-molecule definition for population analysis when it comes to OS prediction with computational methods, but to figure out a scheme that overcomes population analysis and fits more faithfully with the concept of OS. 

Oddly, little attention has been paid to the few computational schemes going beyond the use of conventional population analyses [22,23,24,25,26,27,28,29], while most of them render a very good mapping with the revised IUPAC definition of OS. These OS-oriented methods have in common the fact that they treat electrons individually (or by pairs, in the case of pure singlet states), and then apply one or another strategy to assign individual electrons to atoms, or directly to ligands/molecular fragments. For instance, Ramos-Cordoba et al. [29] introduced some years ago a scheme that is formally applicable on equal footing to any molecular system and wavefunction (e.g., singlet-determinant or correlated, using atomic basis functions or plane-waves). The so-called effective oxidation state (EOS) analysis relies on the abovementioned Mayer’s effective orbitals and their occupation numbers, obtained for all fragments/ligands defined (EFOs). The spin-resolved EFOs are sorted by decreasing occupation number and then individual electrons are assigned to those EFOs with higher occupations, leading to an effective configuration of the atoms/ligands within the molecule, which directly determines their OS. Moreover, the difference between the occupation number of the last occupied and the first unoccupied EFOs is also a pointer of the reliability of the resulting analysis. They are used to derive the index *R*, that ranges from 50% (worst case scenario, with frontier EFOs degenerated in occupancy) to 100%. The larger its value the better the current electron distribution can be pictured into a discrete ionic model. 

Alternatively, there are other approaches that are rooted in the use of localized orbitals [23,24,28]. It is well-known that for single-determinant wavefunctions one can perform unitary transformations to the canonical orbitals of the occupied space that leave the wavefunction unchanged (up to an unimportant phase factor), in such a way that the resulting orbitals appear more localized according to some criterion. Localization schemes are, of course, not unique. Boys [30], Edminston-Ruedenberg (ER) [31], Pipek-Mezey (PM) [32], and most recent improved formulations [27] produce localized orbitals by minimizing some atomic spread functional. A somewhat different strategy is used in the natural bond orbital (NBO) framework [14], aiming at sequentially finding one- and two-center localized orbitals that most closely represent the Lewis structure of the molecule. While the NBOs are not strictly doubly-occupied (i.e., do not span the occupied space), the exact doubly-occupancy can be restored leading to the so-called natural localized molecular orbitals (NLMOs) [33]. 

Thom et al. showed that combining orbital localization schemes with population analysis can lead to an efficient assignment of OS [23]. In their localized orbital bonding analysis (LOBA) scheme, the authors first apply an orbital localization (they seem to prefer ER but also use PM or even NBOs) and then perform a population analysis (Mulliken or Löwdin) on each individual localized orbital to determine in which atom the electron (pair) is localized above a given threshold. The LOBA scheme represents a nice computational mapping of IUPAC’s algorithm of assigning bonds. The scheme proved to be quite robust, but little use has been made of it. Moreover, there are too many choices to be made (localization procedure, population scheme and cutoff threshold) in order to apply it. 

Some of these degrees of freedom are eliminated in the strategy first put forward by Sit et al. [24] and later applied by Vidossich et al. [28]. Sit et al. used maximally localized Wannier functions (MLWFs) [34] as localized orbitals in the framework of plane-waves calculations and obtained the corresponding centroids (Wannier centers). Then, they simply used the position of the centroid to assign the associated electron (pair) to the closest atom. Since MLWFs reduce to the Boys localized orbitals of isolated molecules [34], the same scheme can be applied with computational codes using atomic basis sets. In that case, Vidossich et al. relied on PM localized orbitals, as they allow for σ and π separation. 

Using centroids of localized orbitals to assign OS is very appealing because of its simplicity and, more importantly, because it represents a common framework that can be used indistinctly for molecules and in solid-state. Their use is also gaining recent attention. It has been shown that following the trajectory of the centroids along a chemical reaction allows to recover the curly arrow picture of Robinson [35] from first principles [36,37,38,39]. Previous results indicated that they can be readily used to predict OS, but in our opinion this methodology has not yet been fully tested. 

The aim of this work is to critically assess to which extent the centroids of localized orbitals combined with a distance criterion can be used as an all-purpose scheme to derive OS in molecular systems. We also introduce and test an alternative criterion to assign electrons to centers that makes use of Bader’s topological atoms. atomic basins. We use both NLMO and PM for comparison in selected cases. The performance of these two approaches is compared with that of EOS analysis. 

## 2. Results and Discussion

Let us first consider the series XH_n_, where X = Li to Cl. The results obtained are gathered on Table 1. In this case, there is only one localized orbital whose centroid lies between centers X and H, which of course corresponds to a σ bond between X and H. The remaining localized orbitals correspond to core or lone pairs of X. The position of the centroid determines the ratio *C*_X_ = (*R*_H_ − *R*_X_)/(*R*_H_ + *R*_X_) where *R*_H_ and *R*_X_ indicate the distance between the centroid and the corresponding center. Negative values of the ratio indicate a hydride character and positive values indicate proton. On the other hand, the ratio χ_X_/χ_H_ of Allen’s electronegativity values also discriminate hydride (χ_X_/χ_H_ < 1) from proton (χ_X_/χ_H_ > 1) character according to IUPAC’s ionic approximation. 

One can immediately see that the simplest closest-atom (CA) criterion to assign the electron pair leads in all cases to a formal (−1) charge on the H atom, the only exception being hydrogen fluoride. That is, *C*_X_ values are systematically negative. This result is clearly unsatisfactory, as the simplest H_2_O would be described as a hydride. The same trends are observed using PM or NLMOs. The *C*_X_ values tend to increase (become less negative) along the period and decrease (more negative) along the group, suggesting a relationship with the relative electronegativities of the atoms. In Figure 1, we plot *C*_X_ values obtained with both localization schemes vs. the ratio χ_X_/χ_H_. The correlation is excellent (*r*^2^ = 0.97), but most of the data points associated with proton character according to electronegative are predicted as hydride with the CA criterion. The excellent correlation does indicate that Allen’s electronegativities could be used to correct for the relative atomic size when using the CA criterion. 

However, we believe that introducing electronegativity values in the model would eventually lead to the same problems observed when applying ionic approximation, namely all X-Y bonds would be treated in equal fashion, disregarding their chemical environment. At this point, we decided to explore a different avenue and abandon the CA criterion. Instead, we borrow a key ingredient of Bader’s quantum theory of atoms in molecules (QTAIM), namely the atomic basin. Thus, rather than relying in the distance to a given center, we simply determine which atomic basin the centroid of the bond localized orbital belongs to. This alternative criterion would have some advantages. First of all, the chemical environment of the atoms will be automatically considered, as QTAIM basins are not determined by the nature of the atom but by the electron density, ρ(**r**). Moreover, the relative size of the atomic basins is related to the ionicity of the bond. Secondly, in QTAIM, one should not refer to atoms but to attractors—local maxima of ρ(**r**). There are systems, especially in condensed phase but also in molecules, where an attractor of ρ(**r**) is found away from the nuclear positions. These so-called non-nuclear attractors (NNAs) are a necessary but not sufficient indicator of electride character [40], which could be confirmed by a location of a centroid in its basin (Care must be taken in the case of spurious NNAs, observed for instance in acetylene with some basis sets). Thus, associating the centroid of the localized orbitals to attractor basins readily would incorporate the relative atomic size and environmental effects in the scheme, while introducing little additional arbitrariness. 

This new criterion, henceforth basin-allegiance (BA), is incorporated as follows. First of all, we determine the atomic trust sphere for all atoms of the molecule, as described by Rodriguez et al. [41]. Any point inside a sphere is unambiguously assigned to the corresponding attractor. When a given centroid of localized orbital lies outside the spheres (usually associated to a bond localized orbital), the steepest-ascent path is followed until it reaches a trust sphere. We use both the gradient and the Hessian of the density with a reduced step in all points along the iterative process to ensure a faithful steepest-ascent path. 

The results obtained using the BA criterion are also gathered in Table 1. Now, the hydrogen centers in H_2_O, NH_3_, H_2_S, and HCl are predicted to have proton character as expected. In the case of H_2_S the position of the centroid is extremely close to the bond critical point (*bcp*). It is worth mentioning that we previously observed that the shape of the atomic basins in H_2_S can rather significantly depend on the level of theory used to compute ρ(**r**). Thus, BA behaves essentially in agreement with the electronegativity ratio, with the exception of CH_4_ for which OS have hardly any significance. To further illustrate this point, we depict in Figure 2 the distance of the centroid to the *bcp* versus the electronegativity ratio. In full analogy with Figure 1 for the CA criterion, negative values of the distance indicate that the centroid lies within the H attractor, whereas positive values indicate it belongs to the X atom basin. The correlation is again very good, and in this case all data points (except CH_4_) lie in the right quadrants for hydride and proton character. Thus, introducing the BA criterion clearly improve the results for these systems while adding little extra complexity to the scheme. 

We have also applied EOS analysis to these systems for comparison. In this case, it is the occupation number (*λ*) of the EFOs of H and X that originate from the formal breaking of the H-X bond that lead to hydride (*λ*_H_ > *λ*_X_) or proton (*λ*_H_ < *λ*_X_) character. The ratio (*λ*_H_ − *λ*_X_)/(*λ*_H_ + *λ*_X_) also correlates fairly well with χ_X_/χ_H_, as shown in Figure 3, but not as good as in the previous cases. One can easily see that EOS analysis is also able to discriminate hydride and proton character according to the electronegativity ratio, with the only exception of H_2_S and again CH_4_. The frontier EFOs are almost degenerate in these two examples, indicating a very unpolarized bond, and hence a very poor description of the electron distribution by any ionic model. 

Recently, Postils et al. [5] applied EOS analysis to a set of over a hundred molecular systems. Most of them were included in the IUPAC reports for being either particularly challenging or ambiguous, but the set also included a number of additional examples including π-adducts, high-valent compounds and TM carbenes. The EOS method performed extremely well, even in really intricate bonding situations. In this work, we decided the analyze the performance of the method based upon orbital localization, using both the CA and the just introduced BA criteria for electron assignment, for some particularly relevant examples. When dealing with TM complexes or relatively large systems, it is usually of interest to determine the OS or formal charge of the ligands or molecular fragments as a whole. This is a key point in EOS analysis, where ligands/fragments are defined beforehand. When using centroids of localized orbitals this is not the case, as each individual atom accommodates a number of electrons according to one or another criterion. The formal charge or OS of a given ligand is simply obtained as the sum of the OS over all its atoms. 

Let us consider first the rather simple (CH_3_)_3_NO molecule. In order to fulfill the octet rule for N, a single bond between formal N(+) and O(−) is assumed as the dominant Lewis structure. Then, applying the ionic approximation one assign the two electrons of the σ N-O bond to the O atom, and all N-C bond electrons to the N atom, resulting in oxidation states of (−2) for O, (−1) for N and three CH_3_ units being formally (+1). Both PM and NLMOs point towards the aforementioned Lewis structure, showing a single σ-type N-O localized orbital. No localized orbital corresponding to a N-O π bond is found and, instead, the O moiety bears two p-type lone-pairs. However, as sketched in Figure 4, the centroid of the σ orbital is closer to N than to O atom, and it lies within the atomic basin of N. Thus, for both CA and BA criteria the electron pair corresponding to the N-O σ bond should be assigned to N, leading to a final formal OS of (0) for O and (−3) for N. It is remarkable that the same alternative assignment was obtained using EOS analysis for the same level of theory, and even for a multireference wave function [5]. It appears that the actual electronic structure of this molecule is at odds with the straightforward application of the ionic approximation. 

We have also considered a series of highly-valent TM oxides, namely Ti^(IV)^O_2_, Fe^(VI)^O_4_^2−^, Re^(VII)^O_4_^−^, Os^(VIII)^O_4_, Ir^(IX)^O_4_^+^, and Pt^(X)^O_4_^2+^. EOS analysis performed very well for these systems [5], yielding OS in agreement with the formal values up to (+9) for Ir in IrO_4_^+^ cation [42]. In the case of Pt^(X)^O_4_^2+^, the occupation of the EFO on Pt was too large to be considered empty, so EOS analysis didn’t yield the presumed (+10) oxidation state for this metastable cation [43]. Hence, it is interesting to test the performance of both CA and BA schemes. Since both PM and NLMOs were performing very similarly and we encounter some technical difficulties converging PM localized orbitals for such symmetric systems, we discuss only the results obtained with NLMOs. In all cases, the centroids of localized orbitals corresponding to π bonding between O and the TM were very close to the O atom, indicating almost lone-pair character. The electron pairs under dispute are those of the σ bonds between TM and O. 

In Figure 5 we depict these localized orbitals together with the position of the centroid and the corresponding *bcp* of the density. It can be readily seen that, except for Pt^(X)^O_4_^2+^ cation, the centroid of the σ localized orbital is located between the O atom and the *bcp*. It is closer to the O than to the TM, and therefore both CA and BA criteria yield the expected (−2) OS for the O atoms. This assignment becomes less and less clear cut when going to higher valent compounds. In the case of Ir^(IX)^O_4_^+^ (Figure 5e), while the centroid is still much closer to O (0.931 Å) than to Ir (0.757 Å), it lies very close to the zero-flux surface, ca. 0.003 Å away from the *bcp*. Still, the centroid lies within the atomic basin of O, but one cannot rule out than with a different localization scheme or level of theory the BA assignation could be reversed. 

In the case of Pt^(X)^O_4_^2+^ cation, neither CA or BA criteria predict the presumed (+10) value. First of all, it is worth to point out that the NLMO procedure yielded three very similar Pt-O localized orbitals (Figure 5f) and a fourth one slightly different (Figure 5g). In the former, the centroids were still located closer to O (ca. 0.815 Å) than to Pt (0.894 Å), but already within the basin of Ir, as indicated by the position of the *bcp* in the Figure 5. Therefore, the CA and BA criteria differ in these bonds, yielding different OS assignations. Finally, the last NLMO associated to a Pt-O σ bond is even more polarized towards Pt and its centroid is closer to Pt (and well within Ir basin). All in all, the CA criterion assigns (+8) to Pt, with one the O atoms as (0). With the BA criterion, all O atoms are neutral (0), so Pt is assigned a rather unrealistic (+2) OS. 

The OS assignment in π-adducts can be also problematic in some cases. In this case, the aromaticity of the π-ligand plays a key role, and usually determines the formal charge on the TM. For instance, the C_5_H_5_ moiety is considered as anionic with a formal charge of (−1), thus holding 6π electrons and becoming a Hückel aromatic cyclopentadienyl. In the case of C_7_H_7_ the situation is ambiguous, as both formal charges of (+1) and (−3), accommodating 6π and 10π electrons, are Hückel aromatic and therefore both are plausible. Moreover, Postils et al. recognized that when the π-ligand exhibited some local spin (e.g., triplet character), the OS assignment driven by aromaticity should be (−1), as in the excited-state 8π electron rings become Baird aromatic [44]. This was indeed nicely predicted by EOS analysis [5]. 

Let us see how the CA and BA schemes perform. We have carried out the analysis using NLMOs. for the three π-adducts V(CO)_3_(C_7_H_7_), Mo(C_7_H_7_)(C_5_H_5_), and Mn(C_7_H_7_)_2_. In the first case, there are four localized NLMOs that involve the π system of the ligand, as shown in Figure 6a. The centroids of the localized orbitals lie below the plane, as there is some contribution from the d-orbitals of the TM. In one of them, the contribution from the TM is so relevant that the centroid is pulled from the π-ligand (1.317 Å) on to the V center (0.957 Å), that keeps the electron pair applying both CA and BA criteria. The remaining localized orbitals are associated to the ligand and the final OS assignment is neutral CO ligands, (−1) V and (+1) for the π-ligand. It is worth mentioning that, while (+1) is a plausible assignment for C_7_H_7_, this result differs from that obtained with EOS, leading to a (+3) V and a 10π aromatic (−3) C_7_H_7_ [5]. 

In Mo(C_7_H_7_)(C_5_H_5_), the C_5_H_5_ moiety is readily considered as (−1) anionic for both CA and BA criteria, as could be anticipated. There are up to five localized orbitals with significant contribution from the C_7_H_7_ moiety, depicted in Figure 6b. Some of them exhibit significant contribution from the metal and their centroids appear way below the ring plane. The distances from the centroids to Mo center are 1.717Å, 1.712 Å, 1.264 Å, 1.248 Å, and 1.214 Å. In the last case (Figure 6b, bottom right), the distance between the centroid and the closest C atom of the ring is 1.080 Å, while the zero-flux surface is ca. 1.11Å from the C atom. Thus, all five centroids are closer to the C atoms of the π-ligand and into their atomic basins, leading to 10π electron C_7_H_7_ moiety with a formal OS of (−3), and consequently a Mo center with OS (+4). The same assignment is obtained with EOS analysis in this case [5]. 

The third test system is even more challenging. Here, two nonequivalent C_7_H_7_ rings are bound to a Mn atom. One of the C_7_H_7_ units exhibits noticeable deviation from planarity and its interaction with the TM could be described as η_3_-type, as seen in Figure 7. The whole system is in a doublet state, but there is significant local spin on both the Mn and the ligands. Postils et al. found that, at this level of theory, the system is best described as two triplet C_7_H_7_ units antiferromagnetically coupled to a high-spin d^5^ Mn (+2) center. The EOS assignment of (−1) to the ligands was consistent with 8π Baird aromatic [44] rings in a triplet state (i.e., each ring bears three alpha and five beta π electrons) [5]. 

The alpha and beta NLMOs involving the π-system of the ligands are sketched in Figure 7. For both ligands, there are three alpha and five beta orbitals. In the case of the top C_7_H_7_ ring (Figure 7a), all eight centroids lie much closer to the C atoms and also clearly within their atomic basins, so its OS is (+1), in line with the results of EOS analysis. However, in the bottom ring (actually the one that exhibits η_7_-type coordination with the metal), the centroid of one beta NLMO is closer to the Mn (1.165 Å) than to the C (1.393 Å), and by virtue of the CA criterion is assigned to the metal. Careful inspection shows that the centroid is far from the C atoms but in fact very close to the plane containing the ligand (ca. 0.51 Å). So, considering the C_7_H_7_ ligand as a whole, the centroid could be associated to it. However, careful steepest-ascent path from the centroid leads to the basin of the Mn, so in this case both CA and BA criteria assign only seven π electrons to the second ligand, thus leading to a neutral OS and in consequence a (+1) Mn unit. 

The last set of systems studied are the set of sixteen TM carbenes compiled by Occhipinti et al. [45] and depicted in Figure 8. The set includes four conventional W-based Fischer carbenes (1–4), five Schrock W- and Mo-based catalysts (5–9) and six Ru- and Os-based first- and second generation Grubbs catalysts (10–14) and precatalysts (15–16). We have applied both EOS analysis and CA and BA centroid-based schemes using PM and NLMOs. The key issue is to check the location of the centroids associated to the σ and π TM-carbene bonds (provided they are retrieved by the orbital localization procedure). 

In a Fischer carbene, the σ bond is expected to be polarized towards the carbene, while the π bond should have a large contribution from the TM, leading to a neutral (0) OS of the carbene unit. Again, it is worth remembering that such a picture cannot be derived from IUPAC’s ionic approximation, which considers that all electrons of the bonds must be assigned to either one or another atom. This is the case one would expect for the more nucleophilic Schrock carbenes, with ionic character and a OS of (−2). On the other hand, Grubbs carbenes cannot be easily classified as Fischer or Schrock and, as matter of fact, Occhipinti et al. suggested that a new category of electrophilic Schrock carbenes [45]. 

The OS assignments for the set of TM carbenes are gathered on Table 2. The frontier EFOs and position of relevant orbital centroids and *bcp*s can be found in Appendix A. First of all, all three schemes assign the expected OS to the expectator ligands, namely neutral CO, NHC and phosphine ligands and anionic Cl (−1), tert-butoxide (−1) and phenylimido (−3) ligands (the OS of the TM is entirely determined by that of the ligands and will not be discussed). The only exception is for Grubbs catalyst 10 using NMLO combined with the BA criterion. Here, the σ bond of the phosphine ligand is assigned to the TM, leading to an unrealistic cationic (+2) phosphine. 

According to EOS analysis, the OS of the carbene unit is either neutral (0) or anionic (−2) in all cases. The expected result is obtained for all TM carbenes predefined as Fischer or Schrock, with only one exception. In the case of Grubbs-type carbenes, the low *R* values, near 50% in some cases, indicate that the occupation number of the frontier EFOs on the TM and the carbene are almost degenerate, making the OS assignation uncertain. On the contrary, the OS assignment of the prototypical Schrock carbenes is clearer. In this case, the carbene unit keeps all four electrons of the bonds and becomes formally anionic (−2). These results can be easily visualized in Figure 9. Each molecule is represented by a point in the graph, and its position is determined by the difference of the occupation number of the σ (*x*-axis) and π (*y*-axis) EFOs of the TM and the carbene. Negative values indicate a larger occupation on the carbene. 

The occupation number of the EFOs associated to the σ bond is always larger for the carbene (negative values along *x*-axis), which keeps the σ electron pair. It is the relative occupation numbers of the EFOs associated to the π bonding that ultimately determines the OS. EFO occupation larger for the TM leads to positive values along the *y*-axis. This case corresponds to the typical picture of a Fischer carbene, with a neutral (0) OS. Negative values along the *y*-axis lead to the nucleophilic Schrock character. One can immediately see that the set of Grubbs carbenes are better described as Fisher carbenes (at least from the formal OS point of view), although some of them are right on the frontier. The first- and second-generation Grubbs precatalysts (molecules 15 and 16, respectively) are those with more pronounced Fisher character. 

When using the centroids of localized orbitals, the relative occupation number of the EFOs can be replaced by appropriate distances (see Figure 10a) in order to obtain a graphical representation like that of Figure 9. When using the CA criterion, the σ and π bond distance indices used in Figure 10b are given by the distance between the σ (red dot) and π (green dot) centroids to the midpoint of the TM-carbene bond. In the case of the BA criterion, the reference point becomes roughly the position of the *bcp*. Again, negative distances indicate that the electrons are assigned to the carbene moiety. For instance, Figure 10a illustrates the situation where both CA and BA criteria would be consistent with a Fischer-type neutral carbene. The green dot should be located to the right of the *bcp* in order for both approaches to predict a Schrock-type carbene. Having to consider two centroids at a time increases the risk of getting a different answer from CA and BA criteria, as is indeed the case. 

Unfortunately, the results obtained using centroids of localized orbitals are not satisfactory for several reasons. First of all, one can see in Figure 10b that there are a number of data points corresponding to the Grubbs carbenes for which the σ bond distance index is positive, meaning that the σ electron pair is assigned to the TM rather than to the carbene. At the same time, the π bond distance index is also positive, leading to a hardly acceptable cationic (+2) carbene ligand. This wrong behavior is systematically observed when using the BA criterion, no matter the localized orbitals are PM or NLMO. The CA criterion does a better job for these systems, but the results are somewhat dependent on the orbital localization procedure used, particularly for Grubbs carbenes. With PM localization all data points can be associated to either Fischer or Schrock character, unlike with NLMOs. The behavior of both CA and BA criteria is nevertheless quite good for the prototypical Fischer and Schrock carbenes. 

## 3. Materials and Methods

All calculations were performed with the Gaussian16 package (www.gaussian.com) [46]. Optimized structures and wave functions were determined at the B3LYP/cc-pVTZ level of theory, except for the sets of XH_n_ and TM carbenes, where BP86/def2-TZVP was used instead. PM localized orbitals were obtained using IOp(4/9) = 20212. NLMOs were produced using the NBO6 version [47]. The centroids of localized orbitals were determined by conventional multicenter numerical integration [48] using a 40 × 146 grid per atom with the in-house developed APOST-3D code [49]. The steepest-ascent algorithm from the centroid position to the corresponding attractor was implemented in APOST-3D. EOS analysis was also performed with APOST-3D, using the topological fuzzy Voronoi cells [50] real-space partitioning for the atomic definitions. 

## 4. Conclusions

Finding robust schemes to assign OS from first principles is not a trivial task. The possibility of using centroids of localized orbitals is very attractive, as one could apply the same strategy for molecules and solids on equal footing. Our results, however, indicate that there is no straightforward general use of the centroids to obtain reliable OS. The simplest closest-atom criterion does a good job discriminating Fischer and Schrock carbenes and identifying high-valent species, but fails for the simplest case of H_2_O. An alternative avenue introduced here for the first time consists on determining on which atomic basin each centroid is placed, and distribute the electrons among atoms/attractors accordingly. Such an approach would probably be able to identify electrides and fix the abovementioned hydride/proton issue. However, it does a poor job describing TM carbenes and some metal–ligand interactions. In addition, we have observed that the choice of orbital localization method can have a non-innocent role in the procedure. We have only partially explored the use of PM and NLMO schemes, and it would certainly be necessary to scrutinize more robust alternatives before turning down the use of centroids for OS assignation. In fact, for the present purpose, and in analogy to how EOS analysis is designed, one should probably incorporate the definition of fragments before applying the localization procedure, i.e., on the definition of the orbital spread functional. This work is beyond the scope of this paper. In the meantime, EOS analysis still represents a better approach to obtain OS from first principles. 

## Figures and Tables

**Figure 1 molecules-25-00234-f001:**
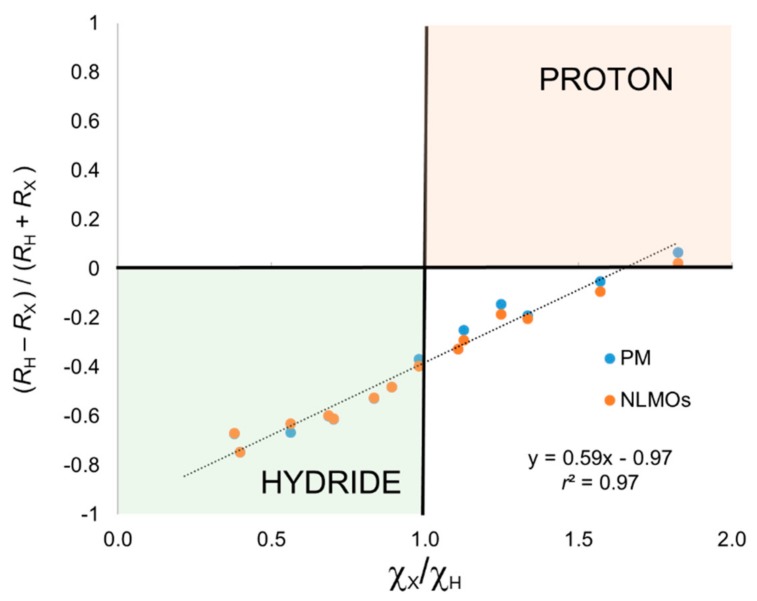
Centroid position (see text) versus electronegativity ratio for the XH_n_ set. OS assignment using closest-atom criterion.

**Figure 2 molecules-25-00234-f002:**
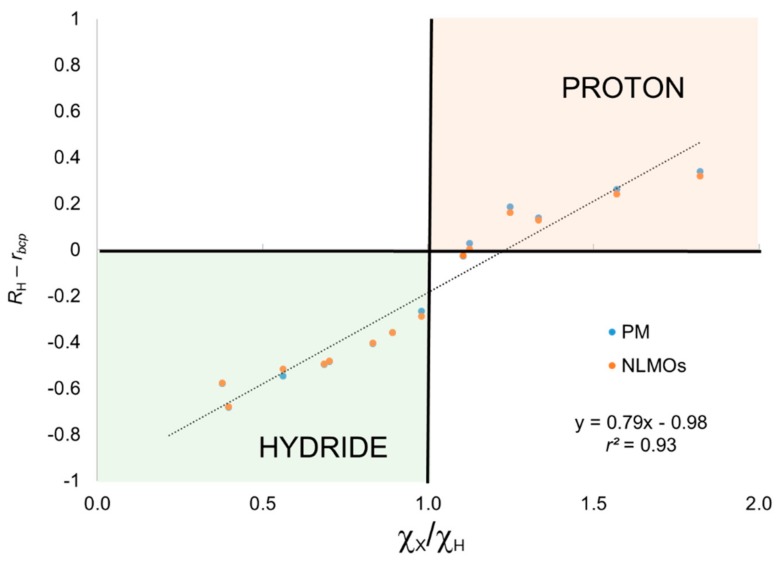
Centroid position relative to the bond critical point (see text) versus electronegativity ratio for the XH_n_ set. OS assignment using basin-allegiance criterion.

**Figure 3 molecules-25-00234-f003:**
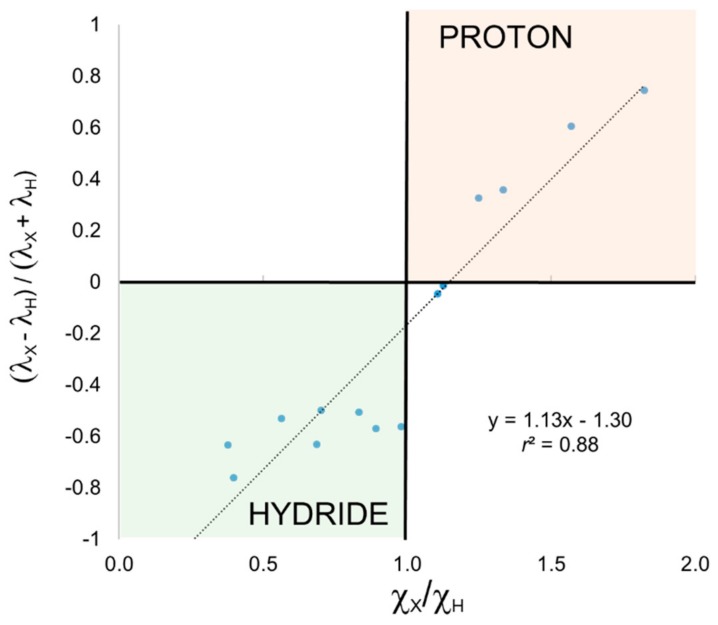
Relative occupation of frontier EFOs versus electronegativity ratio for the XH_n_ set and OS assignment using EOS analysis.

**Figure 4 molecules-25-00234-f004:**
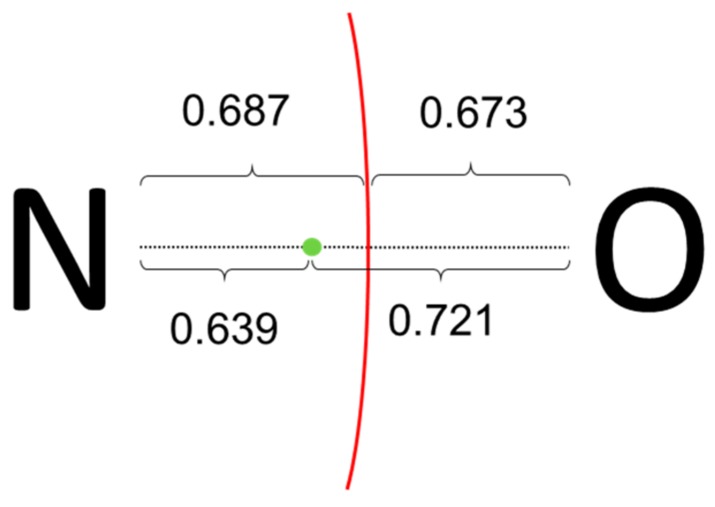
Pictorial representation of the zero-flux surface and position of the centroid of the NLMO σ N-O orbital for (CH3)_3_NO. In the case of PM, the distances from the centroid to N and O are 0.660 and 0.692 (in Å).

**Figure 5 molecules-25-00234-f005:**
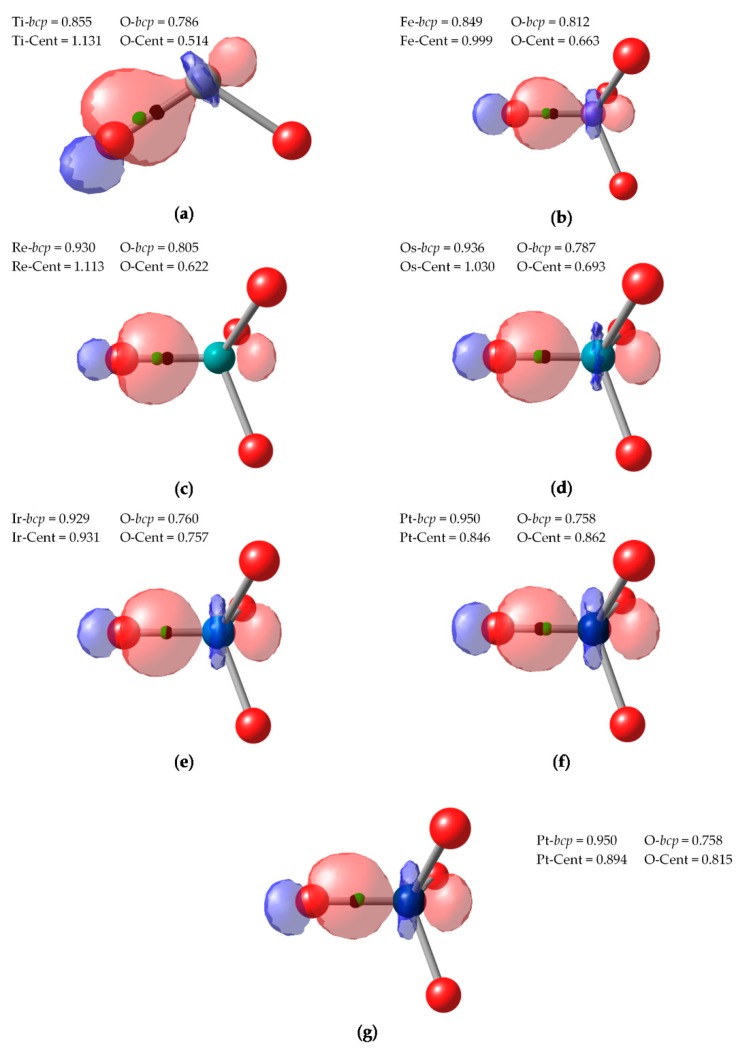
TM-O σ-type NLMO isocontour plot (0.1) for TiO_2_ (**a**), FeO_4_^2−^ (**b**), ReO_4_^−^ (**c**), OsO_4_ (**d**), IrO_4_^+^ (**e**) and PtO_4_^2+^ (**f**,**g**). *bcp* and centroid represented by black and green dots, respectively. (distances in Å).

**Figure 6 molecules-25-00234-f006:**
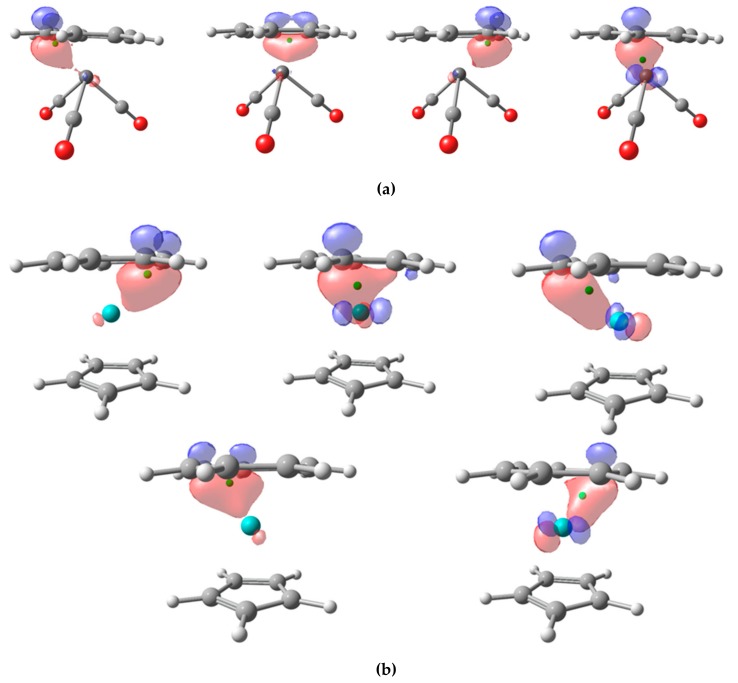
Isocontour plot (0.1) of NLMO localized on the C_7_H_7_ π-ligand for V(CO)_3_(C_7_H_7_) (**a**) and Mo(C_7_H_7_)(C_5_H_5_) (**b**). Orbital centroid represented by green dots.

**Figure 7 molecules-25-00234-f007:**
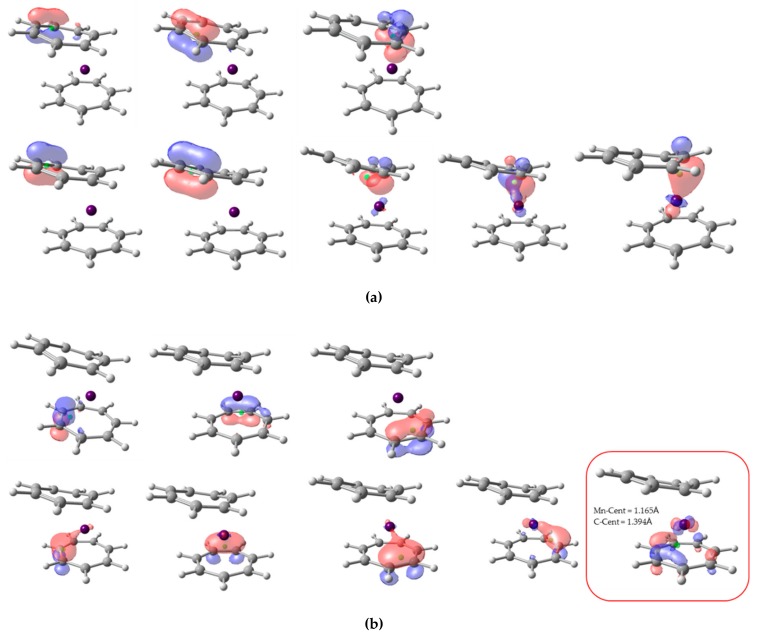
Isocontour plot (0.1) of NLMO localized on top (**a**) and bottom (**b**) π-ligands for Mn(C_7_H_7_)_2_. In each case, top three orbitals correspond to alpha spin, and bottom five to beta spin. Orbital centroid represented by green dots.

**Figure 8 molecules-25-00234-f008:**
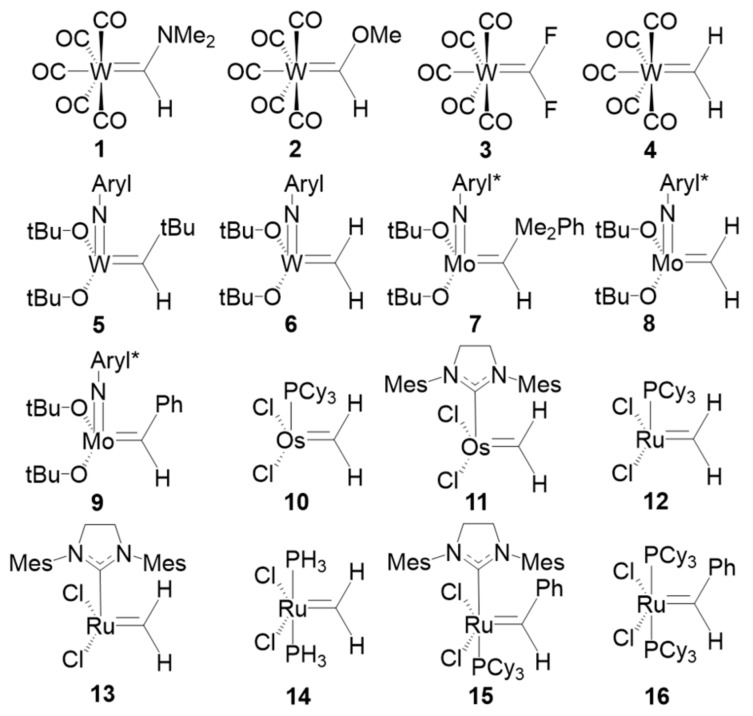
TM carbenes analyzed in this work. Abbreviation: Aryl = 2,6-diisopropylphenyl, Aryl* = 2,6-dimethylphenyl, Cy = cyclohexyl and Mes = mesityl.

**Figure 9 molecules-25-00234-f009:**
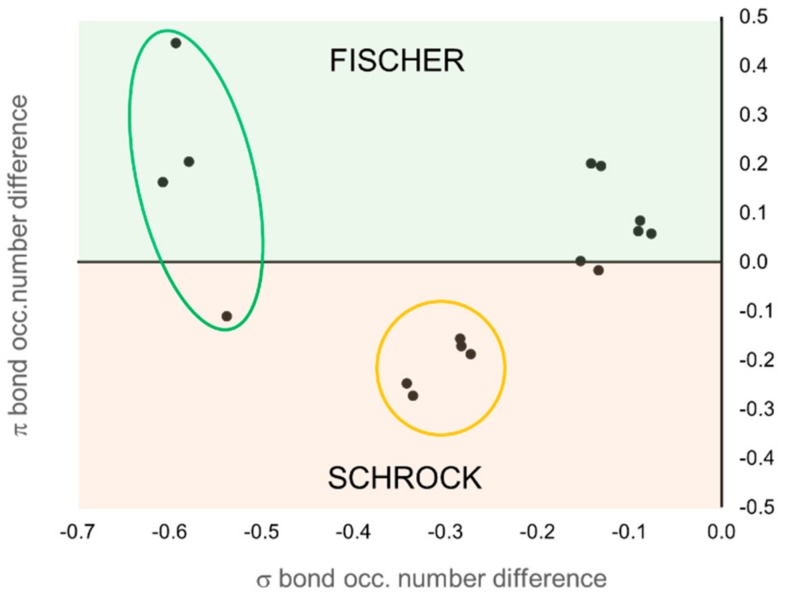
Classification of the TM carbenes according to the relative occupation number of the σ and π EFOs on the TM and the carbene moiety. Data points corresponding to 1–4 (green circle) and 5–9 (orange circle).

**Figure 10 molecules-25-00234-f010:**
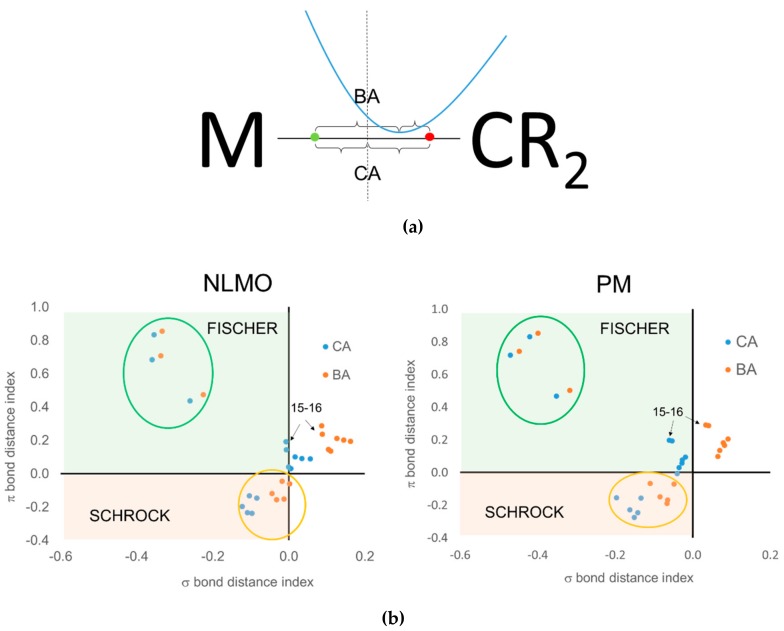
(**a**) Pictorial representation of a Fischer-type carbene, including the centroids of localized σ (red dot) and π (green dot) bond orbitals and relevant distances for CA (midpoint of the bond) and BA (*bcp*) criteria. (**b**) Classification of the TM carbenes according to the distance from the σ and π centroids to bond midpoint (CA) or *bcp* (BA). Data points corresponding to 1–4 (green circle) and 5–9 (orange circle).

**Table 1 molecules-25-00234-t001:** Structural and electronic parameters of the hydrides studied, including Allen’s electronegativity (EN), bond distance, EFO occupancy, distance to the centroid (computed using the PM and NLMO localized orbitals) and distance to the bond critical point (*R*_bcp_-H).

Molecule	Atom	χ_X_/χ_H_	EFO occ. *λ*_X_	*R_X_* (PM)	*R_X_* (NLMO)	*R_bcp_*-H
LiH	Li	0.397	0.112	1.403	1.401	0.886
	H		0.824	0.205	0.206	
BeH_2_	Be	0.685	0.178	1.074	1.072	0.766
	H		0.788	0.270	0.271	
BH_3_	B	0.892	0.195	0.888	0.887	0.669
	H		0.710	0.312	0.312	
CH_4_	C	1.106	0.391	0.727	0.727	0.395
	H		0.429	0.370	0.371	
NH_3_	N	1.333	0.588	0.609	0.615	0.278
	H		0.278	0.416	0.408	
H_2_O	O	1.570	0.729	0.511	0.530	0.200
	H		0.179	0.462	0.442	
HF	F	1.823	0.836	0.435	0.455	0.159
	H		0.122	0.498	0.478	
NaH	Na	0.378	0.170	1.585	1.582	0.892
	H		0.760	0.313	0.316	
MgH_2_	Mg	0.562	0.233	1.427	1.397	0.833
	H		0.760	0.287	0.318	
AlH_3_	Al	0.701	0.249	1.284	1.281	0.792
	H		0.746	0.309	0.312	
SiH_4_	Si	0.833	0.234	1.139	1.136	0.758
	H		0.713	0.353	0.356	
PH_3_	P	0.980	0.183	0.977	0.998	0.720
	H		0.653	0.454	0.433	
H_2_S	S	1.126	0.427	0.845	0.871	0.479
	H		0.439	0.508	0.482	
HCl	Cl	1.247	0.604	0.738	0.765	0.365
	H		0.307	0.553	0.526	

**Table 2 molecules-25-00234-t002:** Formal OS for the carbene moiety (CR_1_R_2_) for the set of TM carbene compounds using centroids of localized orbitals (PM and NLMO) combined with CA and BA criteria and EOS analysis.

Molecule		CA		BA		EOS	*R* (%)
		PM	NLMO	PM	NLMO		
**Fischer**	1	0	0	0	0	0	67.8
	2	0	0	0	0	0	61.2
	3	0	0	0	0	0	59.2
	4	0	0	0	0	−2	56.9
**Schrock**	5	−2	−2	−2	−2	−2	72.0
	6	−2	−2	−2	−2	−2	74.1
	7	−2	−2	−2	−2	−2	64.1
	8	−2	−2	−2	0	−2	66.9
	9	−2	−2	−2	−2	−2	65.7
**Grubbs**	10	−2	+2	+2	+2 ^a^	−2	51.5
	11	0	0	+2	+2	0	50.4
	12	0	+2	+2	+2	0	55.5
	13	0	+2	+2	+2	0	58.1
	14	0	+2	+2	+2	0	55.9
	15	0	0	+2	+2	0	62.4
	16	0	0	+2	+2	0	63.3

^a^ Formal (+2) OS for the phosphine ligand was obtained.

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
