# Peer review of "Can We Safely Obtain Formal Oxidation States from Centroids of Localized Orbitals?"

_molecules, 2020, doi:10.3390/molecules25010234_

Round 1
Reviewer 1 Report
The manuscript of the M. Gimferrer et al., "Can we safely obtain formal oxidation states centroids of localized orbitals?. The authors report a systematic theoretical study and strategic to calculate oxidation states of molecules. In my opinion it is a very interesting work, with a clearly state of the art, using new approach hand it will surely have an important scientific contribution. So, I highly recommend it for publication in molecules.
Author Response
Thank you very much for going throught the manuscript. We are glad you enjoyed the paper in its present form.
Reviewer 2 Report
The manuscript reports nicely about the problems arising with oxidation number and gives suggestions to solve them. Definitions are clearly given and investigations and results are consistently reported.
I can warmheartedly suggest acceptance of the paper for publication.
Author Response
Thank you for your kind words and for going throught the manuscript. We are glad you enjoyed the paper in its present form.
Reviewer 3 Report
The work deals with the problem of assigning the oxidation states from bond analysis performed with theoretical methods. In particular, the analysis refers to centroids of localized orbitals. The need of use of localized orbitals limit the methods to Kohn-Sham approach of DFT. In this work, the level B3LYP/cc-pvtz is used for XH_n molecules while BP86/def2-tzvp is used for transition metal carbenes.
The manuscript is detailed, the presentation is good, is competently written and the
bibliography is exhaustive.
Although I do not expect big changes, it could be intersting to look at the effects of the use of different density functionals (in particular pure GGA, one range separated and,
eventually, with empirical dispersion correction). Moreover, in the case of a significant contribution of static correlation, an unrestricted calculation could lead to different localized orbitals. Also the effect of the basis sets is not considered in this work although mentioned in the introduction.
Finally, is the choice of ECP important in this respect?
Maybe, some discussion on all these aspects coulb be added to the manuscript before
publication.
There is one minor point: the Figure 2 reports for the Y-axis R_H-r_bcp only, is it correct with the possible values that must be in the interval [-1:1]? Please check.
Of course, I recommend publication after minor changes (improvements).
Author Response
Thank you very much for carefully reading the manuscript.
We used different functionals and basis sets for different systems just for better compasion with previous results using EOS analysis. We did check for some systems the effects of basis set and functionals on the shape and centroids of localized orbitals and the results were virtually undistinguishable. Note also ad hoc dispersion corrections such as Grimme's have no effect on the KS orbitals and hence on the localised orbitals.
We do observe significant effects using one or another localization procedure, and this is well discussed in the paper. We believe adding in the mix particular choice of basis set or functional would add unnecessary noise to the message we wish to deliver.
On the other hand, we can confirm figure 2 is correct. As the reviewer correctly points out, the range off the y-axis does not go strictly from -1 to 1; we just kept these values to use tha same framework for figures 1 to 3.
As for the use of ECP, the first rather obvious implication is that all core electrons are automatically assigned to that atomic center.The same occurs with EOS analysis. The lack of core electrons always poses some difficulties with QTAIM analysis. In our case, one just has to be a bit careful when using the BA criterion when choosing the trust sphere.Alternatively one can use one of these techniques to reconstruct the core density from isolated atom all-electron calculations.
Thank you very much afain for your time